# Developing an Updated Strategy for Estimating the Free-Energy Parameters in RNA Duplexes

**DOI:** 10.3390/ijms22189708

**Published:** 2021-09-08

**Authors:** Wayne K. Dawson, Amiu Shino, Gota Kawai, Ella Czarina Morishita

**Affiliations:** 1Veritas In Silico, 1-11-1 Nishigotanda, Shinagawa-ku, Tokyo 141-0031, Japan; ars@vi14si.com; 2Department of Life Science, Faculty of Advanced Engineering, Chiba Institute of Technology, 2-17-1 Tsudanuma, Narashino-shi, Chiba 275-0016, Japan; gota.kawai@p.chibakoudai.jp

**Keywords:** RNA secondary structure, free-energy parameters, Kuhn length, cross-linking entropy, gradient-descent fitting program, genetic algorithm

## Abstract

For the last 20 years, it has been common lore that the free energy of RNA duplexes formed from canonical Watson–Crick base pairs (bps) can be largely approximated with dinucleotide bp parameters and a few simple corrective constants that are duplex independent. Additionally, the standard benchmark set of duplexes used to generate the parameters were GC-rich in the shorter duplexes and AU-rich in the longer duplexes, and the length of the majority of the duplexes ranged between 6 and 8 bps. We were curious if other models would generate similar results and whether adding longer duplexes of 17 bps would affect the conclusions. We developed a gradient-descent fitting program for obtaining free-energy parameters—the changes in Gibbs free energy (ΔG), enthalpy (ΔH), and entropy (ΔS), and the melting temperature (Tm)—directly from the experimental melting curves. Using gradient descent and a genetic algorithm, the duplex melting results were combined with the standard benchmark data to obtain bp parameters. Both the standard (Turner) model and a new model that includes length-dependent terms were tested. Both models could fit the standard benchmark data; however, the new model could handle longer sequences better. We developed an updated strategy for fitting the duplex melting data.

## 1. Introduction

In RNA structure, a stem is a segment of double-stranded RNA (dsRNA) forming a duplex that is typically 3 to 10 bps long. It is the primary information extracted from contact maps [1,2] and forms the scaffolding for all other motifs of RNA structure, particularly the base pairing maps of RNA secondary structure and pseudoknots. From the latter part of the 1950s to the early part of the 1960s, it was quickly recognized from hypochromicity measurements of RNA [3,4,5,6,7,8,9] that there was base pairing [7,8,10,11,12,13], and that these base pairs could result from self-folding of single-stranded RNA (ssRNA)—the concept of RNA secondary structure [5]. Base pairing was used very early in describing the denaturing or melting of RNA duplexes [3,6,14,15,16,17]. Unlike protein secondary structure, which only describes the orientation of adjacent amino acids, RNA secondary structure describes the global base pairing. Fundamental interactions of inter-polymer and intra-polymer chains also formed the bases of much of the work in the 1960s [17,18,19,20,21,22].

A formal description of base pairing parameters began to emerge toward the early 1970s [22,23,24,25,26,27,28,29,30,31,32,33,34,35]. The concept of dinucleotide base pair (2-nt bp) parameters were largely a product of Tinoco’s group in the early 1970s [29]. The 2-nt bp parameters were also called first-neighbor or nearest-neighbor (NN) parameters [29]; these have become the general standard. In that work, nth-neighbor for tri-, tetra-, etc. nucleotide bps were also considered. Owczarzy et al. attempted to measure second-neighbor parameters for canonical Watson–Crick (WC) bps [36,37]. However, in general, the strongest coupling appears to occur between nearest neighboring bases with only a limited degree of coupling between next-nearest-neighbor bases. The precise measurement of 2-nt bp parameters was taken up and improved on mostly by members of the Turner group [38,39,40,41,42,43,44,45]. Similar work was also done with DNA [46,47,48]. The work has been extended to a multitude of motifs beyond RNA/DNA duplexes to include hairpin loops (H-loops) [49,50,51,52,53,54], internal loops and bulges (I-loops) [47,55,56,57], and multibranch loops (M-loops) [58,59,60].

The standard stem motif consists of a dsRNA duplex bound solely by either canonical Watson–Crick (WC) or GU bps, which are often found in the context of other canonical WC bps. The melting of a single duplex composed of contiguous bps forming a single stem motif in the current model can be summarized with the following core equation:(1)$$\Delta G\left(l\right)=\overset{l}{\underset{\mathrm{bp}}{\sum }}{\Delta G}_{\mathrm{bp}}+{\Delta G}_{\mathrm{init}}+{\Delta G}_{\mathrm{sym}}+n_{\mathrm{tAU}}{\Delta G}_{\mathrm{tAU}}$$
where l is the length of the stem, ${\Delta G}_{\mathrm{bp}}$ is the free energy for a given canonical WC or GU 2-nt bp $\begin{array}{c} 5^{\prime }-\mathrm{XY}-3^{\prime }\\ 3^{\prime }-\overset{-}{X}\overset{-}{Y}-3^{\prime } \end{array}$ (where $\overset{-}{X}$ and $\overset{-}{Y}$ reflect a corresponding partner of X and Y, respectively; e.g., $\begin{array}{c} 5^{\prime }-GA-3^{\prime }\\ 3^{\prime }-CU-3^{\prime } \end{array}$), ${\Delta G}_{\mathrm{init}}$ is what has been called the initiation free-energy, ${\Delta G}_{\mathrm{sym}}$ is nonzero when the two sequences forming the duplex are complementary (about 0.5 kcal/mol and independent of sequence length), and $n_{\mathrm{tAU}}$ and ${\Delta G}_{\mathrm{tAU}}$ refer to the number and free-energy correction for terminal AU and GU bps, respectively. For simplicity, 2-nt pairing patterns of the form $\begin{array}{c} 5^{\prime }-\mathrm{XY}-3^{\prime }\\ 3^{\prime }-\overset{-}{X}\overset{-}{Y}-3^{\prime } \end{array}$ will be written with the following shorthand: $\mathrm{XY}\sim \overset{-}{Y}\overset{-}{X}$ or, equivalently, $\mathrm{XY}/\overset{-}{X}\overset{-}{Y}$. Whereas refinements have recently been applied to the original WC 2-nt base pairing parameters [61], the WC duplex parameters measured by Xia [62] have changed only modestly since 1998. The original GU parameters [38] showed more significant changes in later releases. Nevertheless, even in the context of more complete modeling of the thermodynamics of various RNA motifs [63], the fundamental underlying model remains the same.

These base pairing parameters are used in a variety of programs including mfold [64,65,66,67], UNAFold [68], the Vienna package [69,70,71], and a pseudoknot prediction program, vsfold5 [72], and its corresponding suboptimal structure prediction program, vs_subopt [73].

In the development of vsfold5 and vs_subopt, a central concept was that the stem has a particular stiffness that is a function of stem length. The stiffness was defined in terms of the structures’ Kuhn length (ξ) [74,75]. There is also a concept of fraying that is worked into the free-energy model, and fraying is also dependent on sequence length. On the other hand, the Turner model merely applies a constant, ${\Delta G}_{\mathrm{init}}$, to account for unspecified stem formation costs. We are in the process of building a next-generation integrated package based on what we learned from the vsfold5/vs_subopt package that permits variability in the Kuhn length (i.e., stiffness) for different stems in an RNA structure. As some RNA structures can exhibit highly variable degrees of stiffness, we are interested in finding out how the length of the duplex might affect the free energy beyond the sequence-dependent 2-nt base pairing parameters, ${\Delta G}_{\mathrm{tAU}}$ and ${\Delta G}_{\mathrm{init}}$, if at all. To do this, we measured several longer sequences and added parts of that set to the standard benchmark set from Xia et al. [40,61]. From the melting data, we generated base pairing parameters using two different models, the one commonly used to generate the parameters for programs like mfold and the Vienna package and an equation we discuss in this work.

Other approaches that have been shown to work—such as CentroidFold [76], which works with a host of strategies to establish base pairing probabilities; clustering of a Boltzmann-weighted ensemble of RNA secondary structures; a sampling approach; and clustering Sfold [77] and its own maximum expected accuracy estimator, or Pfold [78], which uses stochastic context-free grammars—will not be discussed here. We simply show that a stem model such as the Turner model, although it is generally used and often generates good predictions, is not the only possible model that can be shown to fit the experimental data. Both sequence/structure data and thermodynamic data are fitted using a—gradient-descent (GD) fitting program developed in-house and an approach using a genetic algorithm (GA; based on the Pyevolve package as a driver) to build 2-nt parameters that are native to the cross-linking entropy (CLE) model [50,51,52,53]. For comparison, it is also used to fit duplexes using the Turner model (Equation (1)) with the standard benchmark data used to generate the model parameters.

## 2. Concepts behind the Algorithm

### 2.1. Kuhn Length, Stem Length, and RNA Thermodynamics

In numerous studies of the CLE model, it has been shown that the Kuhn length in folded ssRNA can be largely approximated by assuming that the Kuhn length is proportional to the length of a given stem in an RNA structure. As the length of dsRNA stem is extended, there will be a point where the Kuhn length will essentially reach a maximum. When a complementary sequence of ssRNA is made, the structure becomes dsRNA. Figure 1 compares the change in Kuhn length between three identical transfer RNA (tRNA) structures folded as ssRNA. The same three tRNA structures were combined with their respective complementary sequences and run through a long Monte Carlo simulation, where pentanucleotide correlation was also included. The dsRNA stem remained rather straight and exhibited a vastly different Kuhn length from the corresponding ssRNA structures even though it was the same sequence. Therefore, the context of interaction is just as important as the particular sequence on nucleotides.

### 2.2. The Concept of “Fraying” at the Boundaries of a Contiguous Stem

Even given that we start with a completely contiguous stem, i.e., no mismatches at all, it is unlikely that the Kuhn length is constant throughout the entire length of the stem. Experimentalists have long noted that there appears to be some “fraying” at the boundaries of short oligonucleotide duplex structures, particularly in terminal AU and GU bps [7,35,62,78,79]. Fraying is an effect where instabilities at the boundaries of the stem, probably largely due to interactions with water, penetrate partially into the stem, reducing the stiffness at the boundaries. This is illustrated in Figure 2a–d. Figure 2a shows an unfrayed stem. The stem is stable to all denaturing and disordering interactions. Figure 2b–d reflect the extent to which external interactions can disrupt stacking. One might see this as a kind of penetration depth of solvent interactions along the axis of the stem starting from the ends, as shown in the figure. If the solvent is somewhat denaturing, then the penetration into the stem is likely to be quite deep (Figure 2d). The Kuhn length becomes the average over the length of the stem where the center is the largest and the boundaries the smallest. It is not clear exactly how deep such fraying extends from the ends, but hypochromicity measurements suggest that penetration or fraying could extend as deep as three or four bps [7,10,13,80,81,82], effectively two bps from each end of the structure as in Figure 2c. For normal environments in the cell, extreme fraying in the RNA, as shown in Figure 2d, seems unlikely.

### 2.3. New Free-Energy Model for a Stem

Here, we propose a modified version of Equation (1) that allows accounting for stiffness and fraying of the strands ends:(2)$$\Delta G\left(l\right)=\overset{l}{\underset{\mathrm{bp}}{\sum }}{\Delta G}_{\mathrm{bp}}+{\Delta G}_{\mathrm{sym}}+n_{\mathrm{tAU}}{\Delta G}_{\mathrm{tAU}}+\mathrm{wt}\cdot {\Delta G}_{\mathrm{lcle}}\left(l,{\overset{-}{\xi }}_{\mathrm{stem}}\right)+{\Delta G}_{\mathrm{fray}}\left(l,\xi_{c}\right)$$
where the first three terms are as defined in Equation (1). Although not explicitly written in Equations (1) and (2), temperature (T) dependence is to be assumed for all terms. In the place of ${\Delta G}_{\mathrm{init}}$, two new terms appear at the end of Equation (2). The first term, ${\Delta G}_{\mathrm{lcle}}\left(l,{\overset{-}{\xi }}_{\mathrm{stem}}\right)$, reflects the change in entropy caused by a change in the stiffness as the two independent free single strands combine to form the duplex, and wt is a dimensionless scaling factor. The free strand ssRNA is a very flexible structure, as can be seen in Figure 1 where the loops of the tRNA allow a very compact structure. Depending on the length of the duplex stem, the dsRNA can be far stiffer and becomes even more so as the length of the duplex increase. This is because there is far more order in a double-stranded duplex than in the separate parts and, therefore, a proportional increase in entropy loss [83]. In its simplest form, the free-energy change resulting from the stiffening of a single chain is
(3)$${\Delta G}_{\mathrm{lcle}}\left(l,{\overset{-}{\xi }}_{\mathrm{stem}}\right)=\frac{\left(\gamma +1/2\right)k_{B}T}{D}\int_{1}^{{\overset{-}{\xi }}_{\mathrm{stem}}}\left(\frac{\ln(x)}{1-x}+1\right)\mathrm{dx}$$
where γ is the self-avoiding walk parameter that accounts for the fact that a chain cannot walk back on itself (resulting in a fractal dimension for the system), D is a correction that is proportional to the spatial dimensions of the polymer (D~3), and ${\overset{-}{\xi }}_{\mathrm{stem}}$ can be estimated from the persistence length in a worm-like chain
(4)$${\overset{-}{\xi }}_{\mathrm{stem}}\left(l\right)=\xi_{m}\left\{1-\frac{\xi_{m}}{2l}\left(1-\exp(-2l/\xi_{m})\right)\right\}$$

In Equation (4), $\xi_{m}$ is the maximum Kuhn length for dsRNA (around 200 bps). Note that when l is small, ${\overset{-}{\xi }}_{\mathrm{stem}}\left(l\right)\approx l$, and when l is large, ${\overset{-}{\xi }}_{\mathrm{stem}}\left(l\right)\approx \xi_{m}$. For l > 10 (bp), Equation (3) tends toward a linear increase as proposed by Landau and Lifshitz [84].

The other term that is new in Equation (2), ${\Delta G}_{\mathrm{fray}}\left(l,\xi_{c}\right)$, expresses the fraying of the ends of the duplex, as described in Section 2.2. The exact form of the expression is not known. However, we know from the early studies of Rich et al. [7] that hypochromicity is lower for short oligonucleotide strands, where it was undetectable for strands shorter than seven bps. Therefore, we assume that the intensity of the fraying follows a kind of sigmoidal curve, maximizing at the edges and tapering off deeper inside the duplex. The critical stem length, $\xi_{c}$, represents where the inflection point is. The sharpness of the inflection is defined by the parameter $b_{w}$, and each bp near each end contributes to the entropy loss as a function of depth. A thermodynamic weight $c_{w}$ scales the temperature and inflection dependence. The fraying contribution is an integral of these individual contributions:(5)$${\Delta G}_{\mathrm{fray}}\left(l,\xi_{c}\right)=\frac{c_{w}T}{b_{w}}\left\{b_{w}l+\ln\left[\frac{1+\exp(-b_{w}\xi_{c})}{1+\exp(b_{w}\left(l-\xi_{c}\right))}\right]\right\}$$

Choosing the values $\xi_{c}=4.0$ (bp), $b_{w}=1.0$ (bp^−1^), and $c_{w}=1.0$ (kcal/mol·K·bp), a graph of Equation (5) and its derivative are shown in Figure 3. The derivative of Equation (5) is a sigmoid function indicating the degree of fraying, shown in the green curve in Figure 3. In the first approximation, the sigmoid function should be perfectly symmetric at both ends of the stem, so Figure 3 shows the projection of only one of the ends for clarity. Note also that the variability of the fraying contribution is strongest for very short chains and approaches a constant for longer chains.

Note that the temperature-dependent forms for the free-energy terms in Equation (1) are as follows: ${\Delta G}_{\mathrm{bp}}\left(T\right)={\Delta H}_{\mathrm{bp}}-{T\Delta S}_{\mathrm{bp}}$, ${\Delta G}_{\mathrm{init}}\left(T\right)=-{T\Delta S}_{\mathrm{init}}$, ${\Delta G}_{\mathrm{sym}}\left(T\right)=-{T\Delta S}_{\mathrm{sym}}$, and ${\Delta G}_{\mathrm{tAU}}\left(T\right)=-{T\Delta S}_{\mathrm{tAU}}$, respectively. It is clear, therefore, that ${\Delta G}_{\mathrm{init}}$, ${\Delta G}_{\mathrm{sym}}$, and ${\Delta G}_{\mathrm{tAU}}$ are all derived from entropic phenomena. Indeed, ${\Delta G}_{\mathrm{sym}}$ results from the Gibbs paradox [85] and the correction for concentration when forming a duplex from a self-complementary sequence.

### 2.4. The Duplex Benchmark Set Itself

Here, we call the RNA sequences used to derive the current Turner energy rules for canonical WC bps the “standard benchmark”. This dataset first appeared in Xia et al. [40] and has been revaluated since that time in Chen et al. [45] and Spasic et al. [61]. The lengths of the sequences in the standard benchmark vary from 4 bps to 14 bps. There are only one 9 bp, one 10 bp, and one 14 bp sequence in the dataset. The bp composition is pure GC at 4 bps and gradually shifts toward increasing AU richness as the length of the sequences increase, until the final sequence of 14 bps is pure AU. Largely equal distributions of AU and GC pairs are found for 7 and 8 bps sequences, with some richer in AU. Highly GC-rich sequences only appear for duplexes of 6 bp or less. It is true that a roughly average and generic RNA secondary structure is likely to have stems that range around 6 to 8 bps and most organisms have a roughly equal distribution of A, C, G, and U; hence, the 6 to 8 bp part of the standard benchmark reflects that.

Since most of the sequences in the standard benchmark are 6 and 8 bps long and we are interested in the length dependence of RNA stems, we combined the data from the standard benchmark with the 17 bp sequences from our data. The 17 bp sequences were measured in a physiological salt concentration of 150 mM because our aim was to obtain thermodynamic parameters under conditions that approach the physiological conditions observed for most biological organisms. Chen et al. [86] measured 18 of the duplexes from the standard benchmark under a variety of salt concentrations and derived empirical equations to express melting temperature ($T_{m}$) and ΔG as a function of salt concentration. Both sets contain canonical WC bp patterns. Therefore, we recalibrated the standard benchmark using the reported empirical equations. We also directly recalibrated the data in Chen et al. [86] using linear interpolation of their salt-dependent data.

Within the 17 bp sequences, 14 of the 17 bps were identical for all sequences. To minimize redundancies, we selected only 4 sequences from the 17 bp set and added them to the recalibrated standard benchmark. We fit the combined sequences with both the Turner model in Equation (1) and the new model in Equation (2) using a GD strategy to obtain the free-energy parameters.

### 2.5. Gradient Descent

In GD, a proposed function is fitted with respect to various parameters, as follows:(6)$$h_{\theta }\left(x\right)=\theta_{0}+\theta_{1}x_{1}+f\left(\theta_{2},\theta_{3},x_{2},x_{3}\right)\cdots$$
where $\theta_{0},\ldots ,\theta_{j},\ldots ,\theta_{n}$ is the list of parameters to be fitted (n in total), and $x=\left[x_{0},\ldots ,x_{k},\ldots ,x_{p}\right]$ is a list of various physical conditions associated with a particular piece of data; e.g., $x_{k}$ represents the free energy of a 2-nt bp aa~uu. Here, $f\left(\theta_{2},\theta_{3},x_{2},x_{3}\right)$ represents a function that has parameters $\theta_{2}$ and $\theta_{3}$ along with some associated data properties $x_{2}$ and $x_{3}$. Equation (6) is meant to express the same value as an observable parameter y; in this case, y is the free energy of a specified duplex at 37 °C. Given m independent measurements $y=\left[y_{0},\ldots ,y_{i},\ldots ,y_{m}\right]$, and m corresponding sets of parameters $x_{0},\ldots x_{i},\ldots ,x_{m}$, we built a cost function
(7)$$J\left(\theta \right)=\frac{1}{2m}\overset{m}{\underset{i=1}{\sum }}{\left(h_{\theta }\left(x_{i}\right)-y_{i}\right)}^{2}$$
where i represents a particular piece of experimental data and m is the total number of data points. For data point i, the proposed scalar function $h_{\theta }$ containing the specific physical condition list, $x_{i}={\left[x_{0},\ldots ,x_{p}\right]}_{i}$, is compared with a scalar measured property $y_{i}$; e.g., the measured free energy at 37 °C of sample i. The parameter $\theta_{j}$ is obtained by evaluating
(8)$$\theta_{j}^{\prime }=\theta_{j}-\frac{\alpha_{j}}{m}\overset{m}{\underset{i=1}{\sum }}\left(h_{\theta }\left(x_{i}\right)-y_{i}\right)\frac{\partial h_{\theta }\left(x_{i}\right)}{\partial \theta_{j}}$$
where $\theta_{j}^{\prime }$ becomes the updated parameter $\theta_{j}$ and $\alpha_{j}$ defines the desired learning rate [87], which, in these problems, is around a few parts per hundred. Since the number of parameters is relatively finite, we chose to define the learning rate, $\alpha_{j}$, as an independent variable for each parameter $\theta_{j}$. This allowed us to examine the convergence of individual parameters. In the proposed function $h_{\theta }\left(x_{i}\right)$, $\partial h_{\theta }\left(x_{i}\right)/\partial \theta_{0}=1$, $\partial h_{\theta }\left(x_{i}\right)/\partial \theta_{1}=x_{1}$, $\partial h_{\theta }\left(x_{i}\right)/\partial \theta_{2}=\partial f\left(\theta_{2},\theta_{3},x_{2},x_{3}\right)/\partial \theta_{2}$, and $\partial h_{\theta }\left(x_{i}\right)/\partial \theta_{3}=\partial f\left(\theta_{2},\theta_{3},x_{2},x_{3}\right)/\partial \theta_{3}$. One then proceeds to update all n parameters of $\theta_{j}$ using the m items in the reference data with each iteration. The functions typically approached convergence after a few thousand steps but were carried out to 60,000 iterations.

### 2.6. The Genetic Algorithm

The genetic algorithm has been explained in the literature [87]. The current scoring function was based on four criteria. In these tests, the comparison was based on the difference between the reference structure and what is ultimately predicted by using the dynamic programming algorithm with the given input sequence. As with the previous discussion of GD, we assume m is the number of data points. We also assume the same physical condition list $x_{i}={\left[x_{0},\ldots ,x_{p}\right]}_{i}$, test function $h_{\theta }$, and scalar properties $y_{i}$.

The first criterion is the number of base pairs computed correctly with respect to the reference structure
(9)$${\delta S}_{\mathrm{bp}}=\overset{m}{\underset{i=1}{\sum }}{\left(n_{i,\mathrm{bp}}^{\mathrm{ref}}-n_{i,\mathrm{match}}^{\mathrm{pred}}\right)}^{2}$$
where $n_{i,\mathrm{bp}}^{\mathrm{ref}}$ is the number of bp in the reference structure i, and $n_{i,\mathrm{match}}^{\mathrm{pred}}$ is the number of bp that match between the predicted structure i and the corresponding reference structure.

The second criterion examines the match of the stems, where the tail of the stem from both the reference structure and the predicted structure must match. This is then weighted by the stem length itself. Hence, a poor match between the predicted stems and the reference or a distorted predicted stem will yield an unfavorable (positive) score,
(10)$${\delta S}_{\mathrm{stem}}=\sum_{i=1}^{m}{\left(s_{i,\mathrm{stem}}^{\mathrm{ref}}-s_{i,\mathrm{match}}^{\mathrm{pred}}\right)}^{2},$$
where $s_{i,\mathrm{stem}}^{\mathrm{ref}}$ is the same as $n_{i,\mathrm{bp}}^{\mathrm{ref}}$ but is defined as the number of reference stems in structure i multiplied by the respective stem length, and $s_{i,\mathrm{match}}^{\mathrm{pred}}$ is all the corresponding cases where the predicted structure’s stems match the reference stems.

The third criterion examines the self-consistency between the predicted structure and the reference structure in terms of the free energy,
(11)$${\delta V}_{\mathrm{sc}}=\sum_{i=1}^{m}{\left(V_{i}^{\mathrm{ref}}-V_{i}^{\mathrm{pred}}\right)}^{2},$$
where $V_{i}^{\mathrm{ref}}$ is the calculated free energy of the reference structure i and $V_{i}^{\mathrm{pred}}$ is the calculated free energy of the predicted structure.

Finally, when the data is available, for a fourth criterion, we compare the experimentally obtained free energy of structure i and the predicted free energy,
(12)$${\delta E}_{\mathrm{xpt}}=\overset{m}{\underset{i=1}{\sum }}{\left(E_{i,\mathrm{xpt}}^{\mathrm{ref}}-E_{i,\mathrm{calc}}^{\mathrm{pred}}\right)}^{2}$$
where $E_{i,\mathrm{xpt}}^{\mathrm{ref}}$ is the experimentally measured reference structure and $E_{i,\mathrm{calc}}^{\mathrm{pred}}=V_{i}^{\mathrm{pred}}$.

To compute the score, we define a weighted variance expression,
(13)$$\delta S=\left(w_{\mathrm{bp}}{\delta S}_{\mathrm{bp}}+w_{\mathrm{stem}}{\delta S}_{\mathrm{stem}}+w_{\mathrm{sc}}{\delta V}_{\mathrm{sc}}+w_{\mathrm{xpt}}{\delta E}_{\mathrm{xpt}}\right)/m^{2},$$
where $w_{\mathrm{bp}}{\delta S}_{\mathrm{bp}}$ is the weighted variance of bp matches, $w_{\mathrm{stem}}{\delta S}_{\mathrm{stem}}$ is the weighted stem variance, $w_{\mathrm{sc}}{\delta V}_{\mathrm{sc}}$ is the weighted variance of the calculated reference structures and predicted structures, and $w_{\mathrm{xpt}}{\delta E}_{\mathrm{xpt}}$ is the weighted variance between the observed reference free-energy and the calculated prediction. Finally, the score becomes
(14)score=100exp(−δS)
where a score of 100 would be a perfect score.

## 3. Results and Discussion

To check the basic concepts put forward in the previous section, we fit the standard benchmark (found in [40]) under the condition of 1M salt with both Equations (1) and (2) using the GD tools we developed. The resulting free-energy parameters from fitting the standard benchmark using Equation (1) are shown in Table 1 in column 2 and are compared with those reported in the most recent update [61] in column 4 of the table. The fitted parameters clearly agree within the experimental error bars (column 4) and largely within the estimated error bars (column 3). The predicted free energy of the duplex is plotted against the experimentally evaluated free energy in Figure 4, showing a reasonable match. The inset in Figure 4 reports the chi-squared and residuals of the fit. Hence, the methodology used here reproduces the parameters reported in the literature within the experimental error. Additional details are shown in the Appendix A, Appendix A.

It is important to recognize that Equation (2) must be evaluated in the context of its concepts just as Equation (1) was. Our goal is to understand the concept of a stem and then address the issue of how we build a prediction scheme around it. Therefore, at this stage, we simply ask what happens if we replace Equation (1) with Equation (2)? Can we achieve a similar fit using such a model?

The model in Equation (2) attempts to incorporate additional physical properties such as stiffness and fraying into the characteristic behavior of a generic stem (here an RNA duplex of a given length and bp composition). It is clear from Equation (1) that some kind of positive corrective term is needed to fit the set of duplexes. The constant is independent of the length of the stem and was defined as the initiation free-energy (${\Delta G}_{\mathrm{init}}$). In Equation (2), ${\Delta G}_{\mathrm{init}}$ was suggested in the local cross-linking entropy as accounting for the increasing stiffness of the stem as it becomes longer (${\Delta G}_{\mathrm{lcle}}$) and as correcting for the fraying of stems (${\Delta G}_{\mathrm{fraying}}$), which is particularly important in correcting the free energy of short duplexes. For sequences of similar lengths, it would be reasonable to surmise that ${\Delta G}_{\mathrm{init}}\approx {\Delta G}_{\mathrm{lcle}}+{\Delta G}_{\mathrm{fraying}}$.

Where Equation (2) becomes particularly meaningful is in comparing situations as suggested in Figure 1. The dsRNA structure exhibits a large local-entropic cost because of the long, relatively straight stem that forms along a quasi-one-dimensional axis of the two strands of RNA. The comparatively smaller tRNAs shown in the figure would require far less correction from ${\Delta G}_{\mathrm{lcle}}$, and most of the contribution would tend to come from the fraying of the short stems in the structure. It seems quite doubtful that the ${\Delta G}_{\mathrm{lcle}}$ contributions are in any way similar for these two very different structures in Figure 1, even if we add the four initiation free-energy constants for each tRNA together—it is arguable that this initiation free-energy is absorbed into the so-called “penalties” that are used to calculate the loops in RNA secondary structure calculations. On the other hand, ${\Delta G}_{\mathrm{lcle}}$ is a function of the Kuhn length, which was provisionally defined as proportional to the length of the stem. In that sense, there is also proportionality, but some mitochondrial tRNAs have missing D- or T-loops. What then? In fact, for that whole 212 bp stem, the rules of Equation (1) assert that only one such ${\Delta G}_{\mathrm{init}}$ may be applied.

To explore these questions further, we also fitted the standard benchmark using Equation (2) with the GD tools we developed, assuming the same 1 M salt and 100 micromolar concentrations reported in the experimental conditions. We provisionally kept all other assumptions the same—namely, the 2-nt bp evaluation, terminal AU contribution, and symmetry corrections for self-complementary sequences. In place of the ${\Delta G}_{\mathrm{init}}$ constant, we included the Kuhn length corrections in Equations (3) and (4) and the fraying corrections in Equation (5). The results are also shown in Table 1 (column 5) and in the Appendix A (Appendix A), and the observed and calculated free energies are also plotted in Figure 4, with the results of the fit shown in the inset. The base pairing parameters all appear to be slightly downshifted but, in other respects, they appear rather similar. The fit is only slightly less favorable in its chi-squared and residuals.

The employment of Equations (3) and (5) in the place of ${\Delta G}_{\mathrm{init}}$ resulted in a similar chi-squared and residual as Equation (1). Therefore, this stem model is able to work with the experimental data with largely the same degree of reliability. Context dependence—other than 2-nt bp formalism—is not considered in either model. The free energy is assessed as the sum of these generic stem features, which are entropic in character, and a purely associative combination of 2-nt bps.

We then turned to examining how Equations (1) and (2) would perform if we combined our experimental data for 17 bp duplexes with the standard benchmark. To do this, we were forced to consider that $T_{m}$ was measured in 2.5 micromolar conditions with 150 millimolar salt. Chen et al. [45] proposed a way to estimate $T_{m}$ and ΔG when the concentration of salt is changed from 150 mM to 1 M salt; however, we found that their predicted values did not agree with any salt conditions we tested for 17 bp and 20 bp sequences. We note that the Owczarzy equation does not take into account any length dependence and the SantaLucia equation, although it does, is largely there to calculate an average enthalpy. However, since Chen et al. evaluated 18 duplexes that are in the standard benchmark set, there would be more agreement using these corrections on the standard benchmarks, which it was designed for. Since the range of most of the sequences in the standard benchmark is between 6 and 8 bps, they cover the same sequence length as that for which Chen et al. designed this strategy. We therefore recalibrated the standard benchmarks to 150 mM salt and took into account the different solute concentration conditions; 100 µM vs. 2.5 µM. We introduced four sequences from our 17 bp collection and mixed them with the full standard benchmark. We also attempted to fit one such sequence along with the 18 standard benchmark sequences reported in Chen et al. with a directly interpolated estimate of the free energy and $T_{m}$ corrections.

The results of the fitting Equation (1) and Equation (2) are shown in Table 2 and Figure 5 and in the Appendix A (Appendix A). The chi-squared and residuals are indicated in the inset of Figure 5. Both results have a higher chi-square and residual than found when fitting the standard benchmark in 1M salt conditions. However, the new model has a significantly better agreement with the experimental data. This strongly suggests that length dependence is important in computing stems, as we originally proposed. Indeed, we should be taking into account the stiffness of the RNA when calculating stem structures. There is clearly room for improvement. We are currently working on the details of the stem, and this current rendition is not the final product; it is a conceptualization of the broad issues that remain when predicting RNA structure from thermodynamics, a process that is fraught with difficulties that evolutionary methods of sequence homology are less subject to.

The advantage of using ${\Delta G}_{\mathrm{fray}}\left(l,T\right)$ is that it is length-dependent; so, in principle, this correction can be applied to stems that are even shorter than four bps. It is clear from the general form of the internal loop data that a constant penalty of approximately 4 kcal/mol·K (very close to ${\Delta G}_{\mathrm{init}}$) is also applied. It is known that shifting the internal loop toward the boundaries of the stem tends to increase the instability, i.e., most likely, this fraying tendency is increased.

Figure 2 helps explain why assigning a mere constant in the duplex calculation parameter set produced largely acceptable results; the standard benchmark starts at 4 bps, and the longest sequence was 14 bps. The current dataset is only marginally able to test this hypothesis because the shortest stem length is still four bps long. More tests will be needed to establish the extent to which models, such as that expressed in Equation (2), improve prediction. We are currently testing such models for more complex stems—stems that contain extensive interior loop patterns—using concepts that can be deduced or extrapolated from the stem length-dependent Equations (3)–(5).

The length-dependent corrections that account for stiffness and corrections for fraying required the introduction of only three additional parameters and produced a modest improvement in the fit, as can be seen in the inset in Figure 5. Whereas the fit to the standard benchmark showed no improvement, there was clearly a detectable improvement when other much longer sequences were added. Even in the case of fitting the standard benchmark alone, it is important to emphasize that the models are quite different. Therefore, this shows that there are alternative models for a stem that are just as valid.

In addition to the GD method, we also attempted fitting the standard benchmark using the GA, employing the package drive Pyevolve. The GA works from a different philosophy from GD. As the GA works toward fitness, which is defined by the scoring function in Equation (14) in this case, it is possible to choose a variety of criteria to establish the goodness of a fit. For example, more than one RNA structure might have the same free energy. A fit using only GD can only say that the free energy is a match; however, the GA can contain a score that indicates whether the target structure was found. Then, the score is optimized only when both the target free-energy and the structure are a match. Moreover, the GA has a parallel nature to its search [87]. For example, if the parameters are not so single-valued, solutions from the GA will tend to cluster around multiple regions of higher quality in the scoring landscape. Hence, the GA can find local minima and maxima in the landscape. The GA is fitness driven; it does not depend on evaluating the derivatives or various functions and parameters in the test equation. Only the scoring function matters. Blanket coverage of the parameter space is initially applied. After running a generation, the parameter combinations that yield the fitter results are retained while the poorer ones are gradually suppressed. This results in the hill climbing feature of the GA.

Hence, an important aspect is the definition of the scoring function and how much weight is used for each consideration of the score. We used matching base pairs, stems, and free energy (both calculated for the given structure and observed) as the criteria, shown in Equations (9)–(14). In general, we put the most weight on it matching the experimental free-energy value ${\delta E}_{\mathrm{xpt}}$ and the base pairs ${\delta S}_{\mathrm{bp}}$. An example of the current scoring in the first 20 generations is shown in Figure 6. The blue curve at the top is the maximum score, but we also calculated the average score (purple). The purpose of the green line is to outline the boundaries where most of the scores are found and is defined as 2*average—maximum score. For generation 0, the score is out of range (around 76 in this instance). The scores ranged between 20 and 99 for generation zero. The lower bound of the green line moves up rather rapidly and, from roughly generation 7, fluctuates with the average score. The average score does not necessarily get progressively better with each generation; however, the maximum score does. As a result, there is a gradual improvement with each generation in the maximum score. An important feature of the GA is its so-called “hill climbing” abilities [87].

We see that the advantage of the GA approach over GD is that we obtain some picture of other minima with the landscape. When using GD, one implicitly assumes that there is one and only one solution, whereas the GA need not assume that, though it also aims to achieve convergence. It does appear that there are multiple threads through the solution set. For example, in Table 1 (last column), the parameter found by the GA for AA~UU is significantly different from that of the parameter found by GD; −1.43 vs. −1.17 kcal/mol, respectively. However, if we look at the actual values that Pyevolve selected at each generation for a given individual, it turns out that the program was oscillating between roughly −1.0 and −1.4 kcal/mol. A calculation of the weighted average shows a rather strong shift at various stages of the 100-generation run with a population of 80 (Figure 7). Hence, GD found a compromise solution to the problem that fell between two solutions that GA kept testing. There was also correlation in the attempts made by the GA because there was the tendency that if it attempted the more negative value for AA~UU, it also employed a more positive value for the ${\Delta G}_{\mathrm{lcle}}$ weight. On the other hand, since GD selected a more modest value for AA~UU, it also compromised with a smaller contribution from the ${\Delta G}_{\mathrm{lcle}}$ weight. We are still examining these aspects to understand the landscape better.

## 4. Materials and Methods

To obtain the thermodynamic parameters from experimental data, the melting data were fitted as a function of temperature to a sigmoid function with a linear correction and a small Gaussian function that accounts for some small anomalies around 60 °C. The methods for determining the parameters are explained in detail in the Appendix A. To summarize, after fitting to the sigmoid equation with linear and Gaussian corrections, the corrections were removed from the fit, and the resulting fraction of dsRNA was obtained. This fraction was then fitted with respect to 1/T to generate ΔG and ΔH. The value for ΔG was then adjusted to 37 °C, after correcting for the concentration of solute and salt in the system. These free energies were then fit using either Equation (1) or (2). A schematic is shown in Figure 8a–c.

For the experimental data obtained in this work, the 17 bp oligo-ribonucleotide sequences were ordered from Hokkaido System Science Co., Ltd. (Sapporo, Japan). The UV melting experiments were performed in V-730BIO Spectrophotometer (JASCO Corporation, Tokyo, Japan) using 10 mm path length quartz cuvettes. Other details are explained in the Appendix A.

The remaining free-energy data (at 37 °C) and $T_{m}$ were obtained from Xia et al. [40]: the stems of lengths 4 nt, 6 nt, 8 nt, 10 nt, and 14 nt were all measured at 10^−4^ M substrate concentration and 1 M salt.

To obtain bp parameters, Equation (1) or Equation (2) were fitted using a GD evaluation of the standard benchmark. The data were also fitted using the GA (Figure 6 and Figure 7), obtaining similar results. For the GA, the population was set at 80 and the number of generations set to a maximum of 100.

Figures were made using Pymol (Schrödinger, Tokyo, Japan), Adobe Illustrator (Adobe Inc., San Jose, CA, USA), and gnuplot. All original software code was written in Python 3 and the genetic algorithm code was supported with the Pyevolve package. Gnuplot, python3 and pyevolve are packages that are available to all LINUX users and are developed and maintained by their respective communities.

## 5. Conclusions

In this study, we showed how the integrated fitting package we developed can be merged with our RNA structure program to test different stem models. Here we tested duplexes using the standard benchmark for estimating the free energy of canonical Watson–Crick base pairs and experimental data we measured of longer sequences of 17 base pairs. We used two equations, the standard equation for duplexes and one that we reported here. For the standard benchmark, both expressions worked well. However, when much longer sequences were included in the dataset, the estimates of the standard model were less favorable. The advantage of the model development is that we can test the hypotheses systematically and optimize the parameter sets from first principles, i.e., we can propose a theoretical model and test it. Here, we proposed that stems have the property of fraying at the ends, which adds an entropic cost due to the increased order, that the change in stiffness introduces an entropy correction in the transition between the free strand state, and that the duplex and this strategy appear to be particularly promising for longer sequences.

The package currently employs two independent optimization approaches, GD and the GA. After constructing the appropriate derivatives for functions in a given model, GD is extremely fast, converging close to an asymptotic limit after even a few hundred iterations. However, GD requires considerable care in building datasets and is restricted to examining rather specific themes, such as computing the free energy of RNA duplexes, as presented here. To obtain useful information, GD test sets need to be narrow in scope and aimed at specific physical properties. The GA is far easier to set up and use and is far more adaptable to different datasets. It is also possible to build scoring functions that test a variety of experimentally available features beyond a single parameter like the free energy. However, the GA requires considerably more computational resources. Nevertheless, for that expense, one can see how diverse the solution set turned out to be. Based on an examination of the results from the GA fitting, the solution set was more diverse than the single-valued solution GD suggested. Why? Most likely, it was because the model for the duplex was far too simple and never was reducible to a mere set of 10 dinucleotide base-pair parameters plus a few other parameters or parameterized functions. That said, GD is more likely to generate the best “averages” of such a set of parameters. The GA is, with its hill-climbing abilities, the best way to see the surprises and reveal how single-valued the solution set really is.

Therefore, whereas the model proposed here performed better when additional duplexes of considerably longer length were added to the original dataset and performs as well when challenged with the original dataset, a lot more experimental data are needed in a wide variety of duplex lengths (from 4 to 20 bps at least), base pair arrangements (i.e., context dependence), and physical conditions to achieve a parameter set that approaches a single-valued representation. Such issues remain under investigation. Understanding the fundamentals is essential to moving RNA structure prediction forward from here.

## Figures and Tables

**Figure 1 ijms-22-09708-f001:**
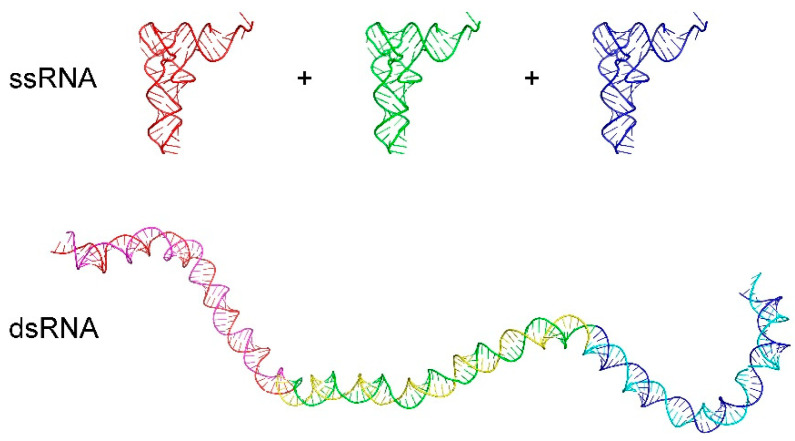
Comparison of the Kuhn length for two types of RNA. Three identical transfer RNA (tRNA, pdb: 1EVV) structures (consisting of folded single-stranded RNA (ssRNA)) are shown at the top and the resulting double-stranded RNA (dsRNA) when all three tRNA sequences are joined with the corresponding complementary sequence are shown at the bottom. Kuhn length is a measure of the tendency for a polymer to remain straight over a given distance. Clearly, dsRNA has a far longer Kuhn length compared to the ssRNA structure for tRNA, and context is a major determinant of stiffness. Images made using Pymol.

**Figure 2 ijms-22-09708-f002:**
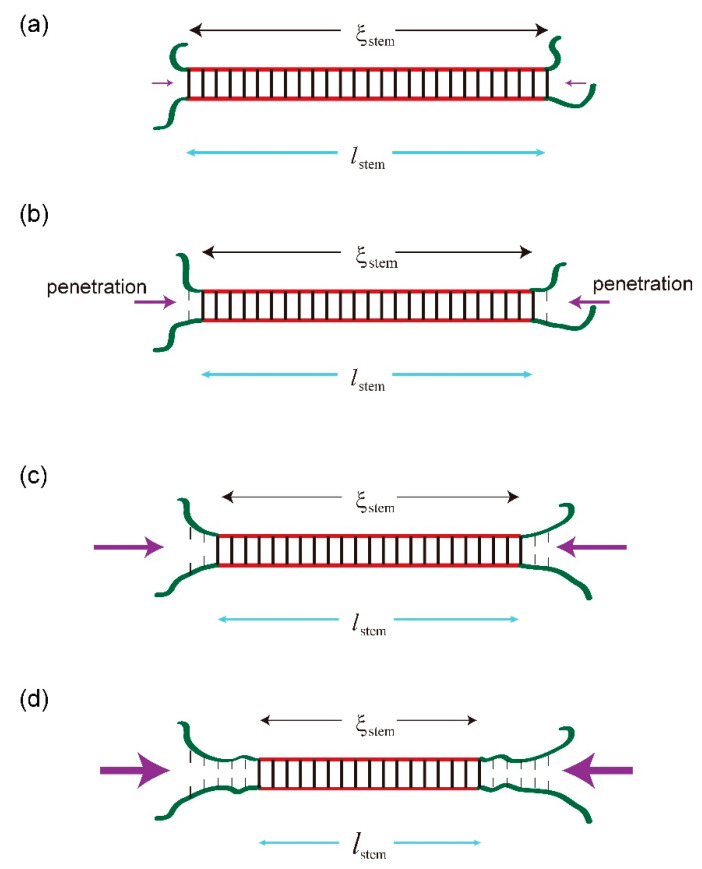
Illustration of fraying in a stem where the stem remains largely ordered, but the ends are slightly disordered: (**a**) a case where there is very little fraying of the stem so the total length of the stem ($l_{\mathrm{stem}}$) and the Kuhn length of the stem ($\xi_{\mathrm{stem}}$) of the stem are proportional; (**b**) a small degree of fraying occurs at the ends of the stem (one base pair (bp)); (**c**) most likely the actual degree of fraying of a typical stem (two bps); (**d**) a more extreme example of fraying. The purple arrows indicate the magnitude of penetration of the solvent into the stem.

**Figure 3 ijms-22-09708-f003:**
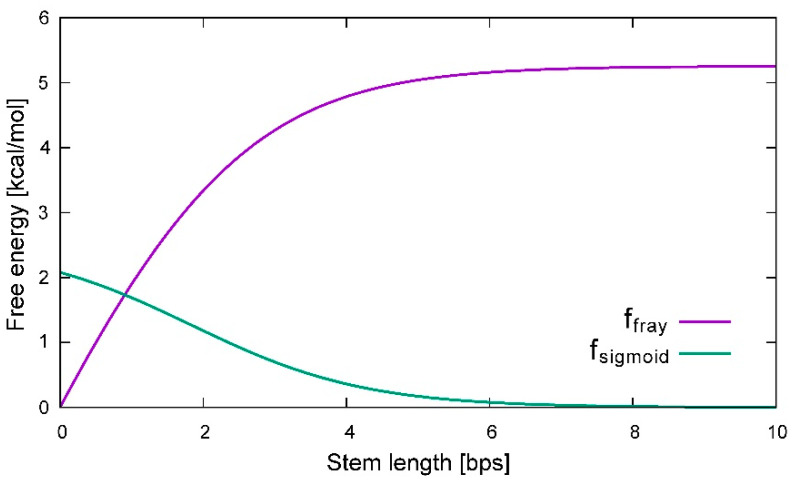
Depiction of fraying corrections as expressed in Equation (5) when the critical length $\xi_{c}=4.0$ (bp), the sharpness of the inflection $b_{w}=1.0$ (bp^−1^), and the thermodynamic weight $c_{w}=1.0$ (kcal/mol·K). For stem lengths longer than 7 bp, the contribution reaches a constant that resembles initiation free energy (${\Delta G}_{\mathrm{init}})$. Graph made using gnuplot.

**Figure 4 ijms-22-09708-f004:**
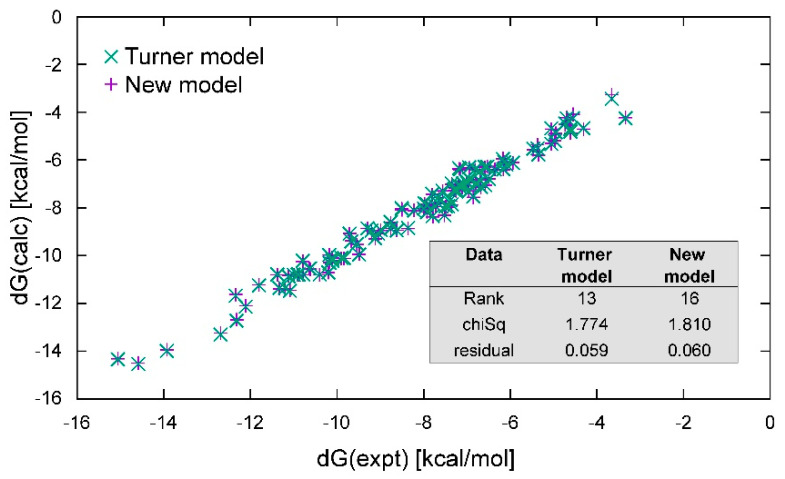
Comparison of fits using Equation (1), the standard Turner model (green x), and Equation (2), an alternative stem model (purple +) for the standard Turner data alone. Here, the results are very similar. The inset table compares the ranks (the number of fit parameters), the chi-squared, and the residuals of the two different models as a result of fitting.

**Figure 5 ijms-22-09708-f005:**
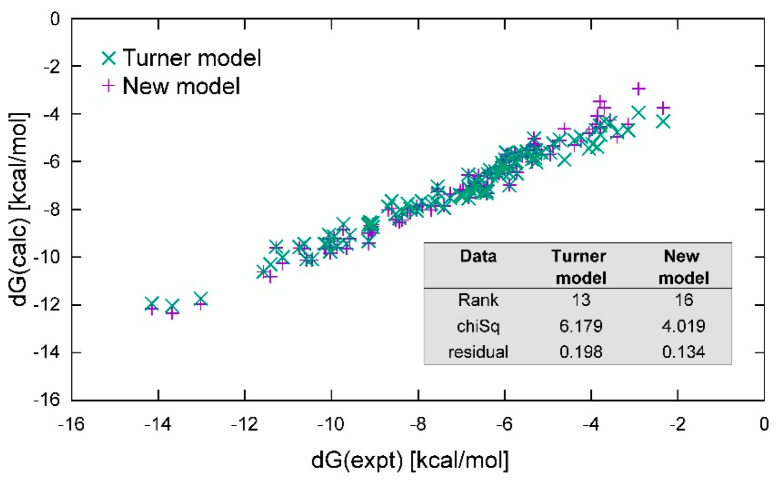
Comparison of fits using Equation (1), the standard Turner model (green x), and Equation (2), an alternative stem model (purple +) for the standard Turner data combined with our data. The recommended correction factors proposed by Chen et al. [86] were used to adjust the thermodynamic parameters at 1.021 M Na^+^ to parameters corresponding to 0.15 M Na^+^, which are the conditions we used to obtain our data. The inset table compares the ranks (the number of fit parameters), the chi-squared, and the residuals of the two different models as a result of fitting.

**Figure 6 ijms-22-09708-f006:**
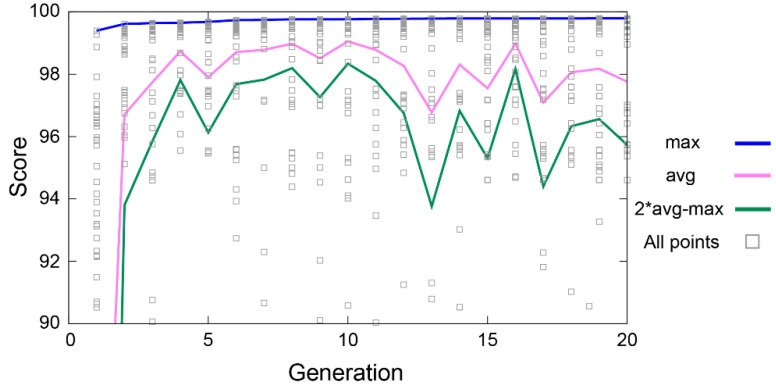
Progress of fitting using the GA using the Pyevolve driver. The blue line shows the maximum score (max), purple the average (avg), green represents 2*avg−max score (i.e., the primary spread), and the gray squares show all the individual scores for each test of the standard benchmark.

**Figure 7 ijms-22-09708-f007:**
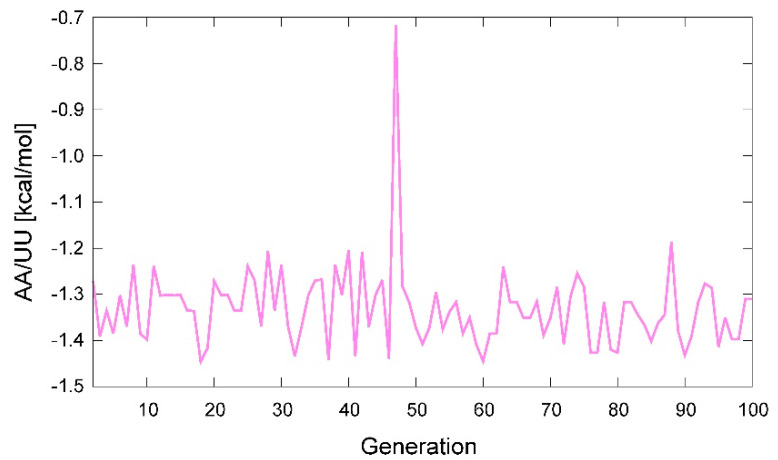
Examination of the weighted average for the parameter AA~UU in Table 1 when using the genetic algorithm (GA). The main oscillation occurs between −1.0 and −1.4 kcal/mol; however, on rare occasions, it even shifts to −0.6 kcal/mol at some locations, such as in the middle of the figure.

**Figure 8 ijms-22-09708-f008:**
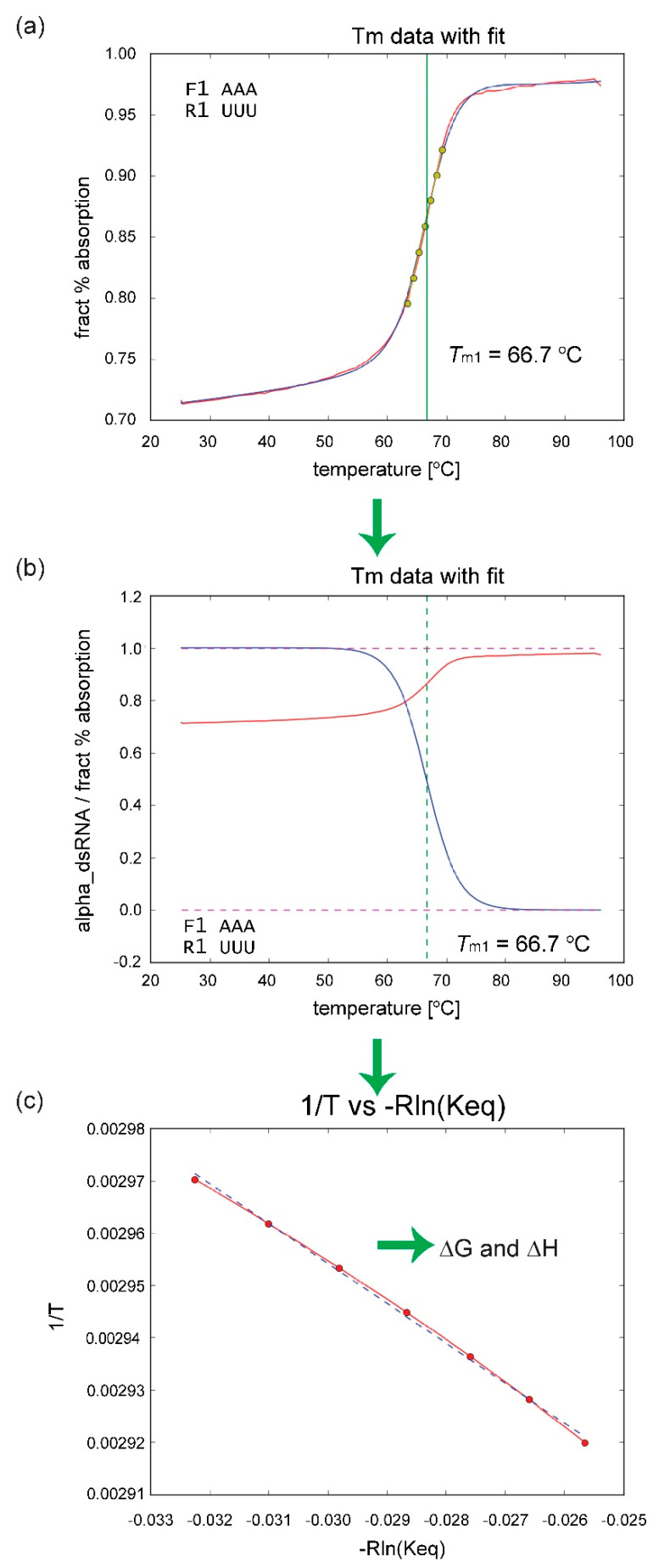
Workflow of how the experimental data were processed and fitted to obtain the free-energy parameters. The details are explained in the Appendix A. (**a**) The data were fit to a function using a GD algorithm. (**b**) Based on the fit, the concentration of dsRNA was deduced. (**c**) Using a fit of the line around $T_{m}$ in (**a**)—the yellow dots—these data were then plotted as $1/T_{m}$ vs. $-R\ln(K_{\mathrm{eq}})$ (as a function of alpha)—from (**b**)—to derive the free-energy parameters: ΔG, ΔH, and ΔS. In (**a**), the red line is the experimentally measured data, the blue line is the resulting fit, the green line marks $T_{m}$, and the yellow points correspond to a linear fit of sigmoid curve for ±3 °C on each side of $T_{m}$. In (**b**), the blue curve is the concentration of dsRNA, the red curve is just a copy of the experimental data from (**a**), and the green line indicates $T_{m}$. In (**c**), the red points and line indicate the experimental data from the seven yellow dots in (**a**) plotted as 1/T vs. $-R\ln(K_{\mathrm{eq}})$, where Keq is derived from (**b**), and the dashed blue line indicates the linear fit of the red dots. See the Appendix A for an explanation of how to obtain Keq.

**Table 1 ijms-22-09708-t001:** Parameters for the standard benchmark [40] and the most recent update [61] using the Turner model, gradient descent (GD), and the genetic algorithm (GA). The symbols and definitions for ${\Delta G}_{\mathrm{bp}}$, ${\Delta G}_{\mathrm{sym}}$, ${\Delta G}_{\mathrm{tAU}}$, ${\Delta G}_{\mathrm{init}}$, ${\Delta G}_{\mathrm{lcle}}$, $c_{w}$, $b_{w}$, and $\xi_{c}$ can found in Equations (1) and (2) and subsequent expressions.

Turner Model			New Model		
Parameters	Fitted Weight ± Error Bars (kcal/mol)		Parameters	Fitted Weight ± Error Bars (kcal/mol)	
	Standard Benchmark	Recent Update		GD	GA
**2-nt bp parameters (ΔG_bp_)**					
AA/UU (aa~uu)	−0.94 ± 0.03	−0.93 ± 0.03	AA/UU (aa~uu)	−1.17 ± 0.03	−1.43 ± 0.03
AC/UG (ac~gu)	−2.26 ± 0.03	−2.24 ± 0.06	AC/UG (ac~gu)	−2.47 ± 0.03	−2.38 ± 0.03
AG/UC (ag~cu)	−2.05 ± 0.03	−2.08 ± 0.06	AG/UC (ag~cu)	−2.26 ± 0.03	−2.35 ± 0.03
AU/UA (au~au)	−1.10 ± 0.04	−1.10 ± 0.08	AU/UA (au~au)	−1.33 ± 0.04	−1.33 ± 0.03
CA/GU (ca~ug)	−2.09 ± 0.03	−2.11 ± 0.07	CA/GU (ca~ug)	−2.29 ± 0.03	−2.28 ± 0.03
CC/GG (cc~gg)	−3.33 ± 0.03	−3.26 ± 0.07	CC/GG (cc~gg)	−3.54 ± 0.03	−3.44 ± 0.03
CG/GC (cg~cg)	−2.29 ± 0.03	−2.36 ± 0.09	CG/GC (cg~cg)	−2.50 ± 0.03	−2.58 ± 0.03
GA/CU (ga~uc)	−2.43 ± 0.03	−2.35 ± 0.06	GA/CU (ga~uc)	−2.64 ± 0.03	−2.66 ± 0.03
GC/CG (gc~gc)	−3.55 ± 0.03	−3.42 ± 0.08	GC/CG (gc~gc)	−3.76 ± 0.03	−3.65 ± 0.03
UA/AU (ua~ua)	−1.36 ± 0.04	−1.33 ± 0.09	UA/AU (ua~ua)	−1.57 ± 0.04	−1.50 ± 0.03
**Sequence-independent parameters**					
ΔG_sym_	0.47 ± 0.02	0.5 **	ΔG_sym_	0.46 ± 0.02	0.37 ± 0.03
ΔG_tAU_	0.38 ± 0.02	0.45 ± 0.04	ΔG_tAU_	0.38 ± 0.02	0.78 ± 0.03
ΔG_init_	4.22 ± 0.02	4.10 ± 0.02	ΔG_lcle_ → wt *	0.60 ± 0.02 *	1.16 ± 0.03 *
			ΔG_fray_ → c_w_ *	2.53 ± 0.02 *	2.40 ± 0.03 *
			ΔG_fray_ → b_w_ *	0.83 ± 0.02 *	0.78 ± 0.03 *
			ΔG_fray_ → ξ_c_ *	1.84 ± 0.02 *	1.72 ± 0.03 *

* Here f(x)→b indicates b is a parameter of the function f(x). ΔG_lcle_ → wt does not have units. The unit for ΔG_fray_ → c_w_ is kcal/mol∙K∙bp, and the unit for both ΔG_fray_ → b_w_ and ΔG_fray_ → ξ_c_ is bp^−1^. ** Not measured in this work.

**Table 2 ijms-22-09708-t002:** Parameter comparison for the standard Turner data combined with our data using the Turner and new models. The symbols and definitions for ${\Delta G}_{\mathrm{bp}}$, ${\Delta G}_{\mathrm{sym}}$, ${\Delta G}_{\mathrm{tAU}}$, ${\Delta G}_{\mathrm{init}}$, ${\Delta G}_{\mathrm{lcle}}$, $c_{w}$, $b_{w}$, and $\xi_{c}$ can found in Equations (1) and (2) and subsequent expressions.

Turner Model		New Model	
Parameters	Fitted Weight ± Error Bars (kcal/mol)	Parameters	Fitted Weight ± Error Bars (kcal/mol)
	Standard Benchmark		GD
AA/UU (aa~uu)	−0.71 ± 0.06	AA/UU (aa~uu)	−1.65 ± 0.05
AC/UG (ac~gu)	−1.78 ± 0.05	AC/UG (ac~gu)	−2.73 ± 0.04
AG/UC (ag~cu)	−1.48 ± 0.05	AG/UC (ag~cu)	−2.48 ± 0.04
AU/UA (au~au)	−0.58 ± 0.06	AU/UA (au~au)	−1.62 ± 0.05
CA/GU (ca~ug)	−1.55 ± 0.05	CA/GU (ca~ug)	−2.51 ± 0.04
CC/GG (cc~gg)	−2.69 ± 0.06	CC/GG (cc~gg)	−3.70 ± 0.05
CG/GC (cg~cg)	−1.56 ± 0.05	CG/GC (cg~cg)	−2.74 ± 0.04
GA/CU (ga~uc)	−2.07 ± 0.05	GA/CU (ga~uc)	−2.94 ± 0.04
GC/CG (gc~gc)	−3.20 ± 0.05	GC/CG (gc~gc)	−4.05 ± 0.04
UA/AU (ua~ua)	−0.95 ± 0.07	UA/AU (ua~ua)	−1.79 ± 0.06
ΔG_sym_	0.39 ± 0.04	ΔG_sym_	0.40 ± 0.03
ΔG_tAU_	0.32 ± 0.04	ΔG_tAU_	0.39 ± 0.03
ΔG_init_	2.56 ± 0.04	ΔG_lcle_ → wt *	2.28 ± 0.03 *
		ΔG_fray_ → c_w_ *	2.79 ± 0.03 *
		ΔG_fray_ → b_w_ *	1.07 ± 0.03 *
		ΔG_fray_ → ξ_c_ *	2.25 ± 0.03 *

* Here f(x)→b indicates b is a parameter of the function f(x). ΔG_lcle_ → wt does not have units. The unit for ΔG_fray_ → c_w_ is kcal/mol∙K∙bp, and the unit for both ΔG_fray_ → b_w_ and ΔG_fray_ → ξ_c_ is bp^−1^.

## Data Availability

The data used in the benchmark are from Xia et al., 1998. The software and data from the 17 bp sequences are available on request. Some of the data are indicated in Appendix A.

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
