# Peer review of "Developing an Updated Strategy for Estimating the Free-Energy Parameters in RNA Duplexes"

_ijms, 2021, doi:10.3390/ijms22189708_

Round 1

Reviewer 1 Report

The manuscript "Developing an updated strategy for the free energy parameters in RNA duplexes" by Dawson et al., is a very sound and well written article. Although the new fitting model they present here does not show any considerable improvement compared to the standard model, it does include interesting new concepts and parameters that expand the understanding of the underlying behaviour of these systems and, in the future, might help to further improve our current theoretical models.

Therefore, my only suggestion before the article is published, is to check or correct some very minor syntax/spelling errors:

  • I would suggest checking the title, something seems to be missing (e.g., Developing an updated strategy for [¿estimating? ¿evaluating?] the free energy parameters in RNA duplexes).
  • Line 274, "in in". Please correct.
  • Lines 616-618, the font used for the years is different.

Author Response

Dear Reviewer,

Because of the format at IJMS, I include the revisions to the paper and supplement as a single file. Changes are indicated in red, or blue in the case of the tables.

Sincerely, 

wd

Reviewer 2 Report

The authors presented originally fitted free energy parameters for RNA duplexes (up to 17 bps) based on formula that take stiffness of the duplexes and fraying of the boundaries of the duplexes into account. In this parameter fitting, gradient descent (GD) and genetic algorithm (GA) are utilized as numerical optimizers. Although improving free energy parameters is one of the important issues in RNA informatics, in current from more description is necessary to improve the readability.

Major points:

-The correspondence between the parameter names used in Tables and those in equations is not clear. E.g. in Table 1, the values for delta_G_lcle wt, delta_G_fray (cw) delta_G_fray (bw) delta_G_fray (c) are shown as fitted values of the new model’s parameters. However, in Eq. 2, 3, and 4, exactly the same notations cannot be found.

- In the results shown in the paper, I cannot find the reason for employing the GA to fit the parameters, since the authors do not show quality measure such as residual for the fitted results by GA and do not compare such quality measure values between the results of GD and GA.

-Reproducibility of the results (software): The authors mentioned in line 460-461 that “we have shown how the integrated fitting package we developed can be merged with our RNA structure program to actually test different stem models”. However, I was not able to find a link to such an integrated fitting package in the paper and supplementary materials. To confirm the reproducibility of the presented fitting results, it is important that readers can utilize source codes, executables or web server that can reproduce the results given in the paper.

The following link gives instruction for authors about software:

https://www.mdpi.com/journal/ijms/instructions#suppmaterials

-Four 17-bps sequences added in this paper: I think the number of additional 17 pbs sequences (= four sequences) is very small compared with the increment in the number of parameters (In the new model proposed in this paper, the number of parameters were increased from thirteen [the standard model] to sixteen [the new model]). What do the authors think about this point?

-Figure captions: In each figure cation, more description is needed. E.g., in Figures 8, 1m, 2m, and 3m, explanations for red solid lines, blue solid lines, and dotted lines should be added to the corresponding captions.

-Captions of tables: No description is found for the asterisks used in the tables.

Minor points:

-Line 318: other 2-nt -> other than 2-nt

-Line 400, the caption of Figure 6: What is the green line? Minimum or the range between the maximum and the average?

-Line 468: “from first principles”: Please explain the “first principles” used here in more details.

-Line 472: Please explain the “some anomalies that remain under investigation” in more details.

Author Response

(The authors gave the same response as above.)

Round 2

Reviewer 2 Report

This paper is much improved from the first version. 

Minor points:

-In Figure 6, "minimum" should be deleted since it  just indicates the position that is located at "average" - ("maximum" - "average"), which corresponds neither to the minimum value of the data nor to the lower part of the spread of the data.

-In Conclusion section: It is better to mention that two sets of parameters were determined by GD and GA. In addition, if possible, describing which one is more reliable is also interesting discussion to readers. 
